# The Amino Acid Permease MoGap1 Regulates TOR Activity and Autophagy in *Magnaporthe oryzae*

**DOI:** 10.3390/ijms232113663

**Published:** 2022-11-07

**Authors:** Changli Huang, Lin Li, Lei Wang, Jiandong Bao, Xiaozhi Zhang, Jiongyi Yan, Jiaqi Wu, Na Cao, Jiaoyu Wang, Lili Zhao, Xiaohong Liu, Xiaoping Yu, Xueming Zhu, Fucheng Lin

**Affiliations:** 1Zhejiang Provincial Key Laboratory of Biometrology and Inspection & Quarantine, College of Life Sciences, China Jiliang University, Hangzhou 310018, China; 2State Key Laboratory for Managing Biotic and Chemical Threats to the Quality and Safety of Agro-Products, Key Laboratory of Biotechnology in Plant Protection, Ministry of Agriculture and Rural Affairs, Institute of Plant Protection and Microbiology, Zhejiang Academy of Agricultural Sciences, Hangzhou 310021, China; 3College of Advanced Agricultural Sciences, Zhejiang Agriculture and Forest University, Hangzhou 310007, China; 4State Key Laboratory of Rice Biology, Institute of Biotechnology, Zhejiang University, Hangzhou 310058, China; 5College of Biotechnology, Tianjin University of Science & Technology, Tianjin 300457, China

**Keywords:** rice blast, gene knockout, TOR, autophagy, amino acid permease

## Abstract

Rice is an important food crop all over the world. It can be infected by the rice blast fungus *Magnaporthe oryzae*, which results in a significant reduction in rice yield. The infection mechanism of *M. oryzae* has been an academic focus for a long time. It has been found that G protein, AMPK, cAMP-PKA, and MPS1-MAPK pathways play different roles in the infection process. Recently, the function of TOR signaling in regulating cell growth and autophagy by receiving nutritional signals generated by plant pathogenic fungi has been demonstrated, but its regulatory mechanism in response to the nutritional signals remains unclear. In this study, a yeast amino acid permease homologue MoGap1 was identified and a knockout mutant of MoGap1 was successfully obtained. Through a phenotypic analysis, a stress analysis, autophagy flux detection, and a TOR activity analysis, we found that the deletion of MoGap1 led to a sporulation reduction as well as increased sensitivity to cell wall stress and carbon source stress in *M. oryzae*. The Δ*Mogap1* mutant showed high sensitivity to the TOR inhibitor rapamycin. A Western blot analysis further confirmed that the TOR activity significantly decreased, which improved the level of autophagy. The results suggested that MoGap1, as an upstream regulator of TOR signaling, regulated autophagy and responded to adversities such as cell wall stress by regulating the TOR activity.

## 1. Introduction

Rice is the third most important food crop in the world after wheat and maize, and more than half of the world’s population depends on it [1]. Rice blast is one of the most serious rice diseases worldwide and is mainly caused by the rice blast fungus *Magnaporthe oryzae*. When the conidia of the fungal asexual generations are formed, their mycelia can invade the leaves, stems, and other parts of rice, forming leaf blast, knot blast, and panicle plague to affect the entire growth cycle of rice. Due to the operability of the molecular inheritance of *M. oryzae*, it has become one of the top ten plant pathogenic fungi [2]. In recent decades, G protein, AMPK, cAMP-PKA, MPS1-MAPK, and other signaling pathways have been well-studied and their involvement in regulating the pathogenic process of *M. oryzae* has been confirmed. In the rice blast fungus, Pmk1 and Mps1-MAP kinases with the TEY motif are also critical for cell growth, osmosis, and infestation [3]. Recently, Qian et al. found that nutritional hunger was a necessary condition for the formation and functioning of the appressoria. In their observations, TOR inhibition reduced the affinity between MoTap42 and MoPpe1 (a protein phosphatase involved in the TOR signaling pathway) and released MoPpe1 to activate the cell wall integrity (CWI) signaling pathway, which has a significant impact on vegetative growth, the formation of fungal conidia, and the virulence of pathogens by mediating the crosstalk between TOR and CWI signaling pathways [4].

Autophagy is a necessary evolutionarily conserved physiological process, which maintains cellular homeostasis and survival by degrading defective proteins, other cellular contents, and substances that cause stress to the cell from the outside [5,6,7]. In response to environmental stresses such as starvation, injury, or reactive oxygen, autophagosomes are formed in cells to wrap the damaged macromolecular substances such as proteins and lipids. After being devoured by lysosomes/the vacuole, these macromolecular substances are decomposed into small molecular substances such as amino acids, nucleotides, and fatty acids for recycling [8,9,10]. Autophagy can be induced by nutritional deprivation and this induction is influenced by the activity of several protein kinases such as those in the TOR signaling pathway. Activated TOR signaling hyperphosphorylates Atg13, thereby preventing the assembly of the Atg1/Atg13 complex and inhibiting autophagy [11,12]. The dephosphorylation of Atg13 after the inactivation of TOR signaling results in the formation of an active complex that contains Atg1, Atg11, and Atg101 as well as the Atg13-Atg17-Atg1 complex, the latter of which is localized to the phagophore assembly site (PAS) and initiates autophagy in *A. thaliana* [13,14,15,16]. Recent studies have shown that TOR protein kinases, as the upstream regulators of autophagy, can regulate membrane tension, carbon source utilization, nitrogen source utilization, and CWI [17]. However, the TOR upstream regulatory mechanism of different response signals remains unclear.

Amino acid transporters, a large family of membrane proteins, are located on the plasma membrane (PM). These proteins play crucial roles in signaling transduction and material transportation. In yeast, Gap1 (general amino acid permease 1) is an amino acid transporter, which transports almost all natural L-amino acids and mediates the amino acid signaling pathway [18]. Under nitrogen sources deprivation conditions, Gap1 was located on the PM surface to increase its permeability and promote amino acid transportation from the outside to the inside of a cell. In yeast, amino acid signaling plays a key role in the regulation of autophagy. When Gap1 transports large amounts of amino acids into the cell, the TORC1 kinase complex is activated, which maintains cell growth and mitosis [19,20,21]. In order to understand the relationship between MoGap1 and TOR signaling in *M. oryzae* and explore the role of MoGap1 in autophagy, this study successfully obtained a *MoGAP1* mutant through a gene knockout strategy. Our results showed that the deletion of MoGap1 increased the sensitivity to cell wall stress and carbon source stress in *M. oryzae*. In addition, the loss of MoGap1 led to a high sensitivity to rapamycin (the TOR inhibitor) and the TOR activity was significantly decreased in Δ*Mogap1*. Moreover, the level of autophagy significantly increased due to the low activity of TOR. These results suggested that MoGap1, as an upstream regulator of TOR signaling, regulated autophagy and responded to cell wall stress by regulating the TOR activity, which provided a potential theoretical basis for analyzing the function of MoGap1 in pathogenic fungi.

## 2. Results

### 2.1. Identification and Knockout of MoGAP1 in M. oryzae

In the yeast *Saccharomyces cerevisiae*, Gap1 acts as an amino acid permease that transports naturally occurring L-amino acids and mediates the amino acid signaling pathway. In order to explore the function of the amino acid permease in *M. oryzae*, we identified the Gap1 homologous proteins in *M. oryzae*, *S. cerevisiae*, *Schizosaccharomyces pombe*, *Colletotrichum truncatum*, and *Fusarium graminearum*. The domain conservation analysis showed that the Gap1 protein contained a Lysp domain and 13 transmembrane regions (Figure 1A,B). An homology analysis sequence alignment found that MoGap1 had a 53.25%, 45.65%, 64.84%, and 62.92% protein identity with the general amino acid permease Gap1 proteins of *S. cerevisiae*, *S. pombe*, *C. truncatum*, and *F. graminearum*, respectively, indicating that Gap1 is highly conserved in fungi (Figure 1C,D). In order to discover the function of MoGap1 in *M. oryzae*, we successfully obtained a *MoGAP1* mutant through a gene knockout strategy, as described by Lu et al. [22]. As shown in Appendix A, the target gene existed in the wild-type strain Guy11, but not in the Δ*Mogap1* mutant; the recombinant fragments only existed in Δ*Mogap1*. In addition, we also detected relative amounts of the *HPH* gene and *MoGAP1* gene in Δ*Mogap1*. In Δ*Mogap1*, the ratio of the *HPH* gene was near 1.0 whereas the ratio of the *MoGAP1* gene was near 0 (Appendix A), indicating that the *MoGAP1* gene was successfully knocked out in the wild-type Guy11.

### 2.2. MoGap1 Regulates the Formation of the Conidia

Δ*Mogap1* and the complementary strain Δ*Mogap1-C* were cultured on complete media (CM) for 9 days, after which the colony morphology was observed. There were no significant differences in the morphology of Guy11, Δ*Mogap1*, and Δ*Mogap1-C* (Figure 2A). In the wild-type Guy11 and the complementary strain, the amount of conidia was up to 2.0 × 10^6^ per milliliter. However, only 6.0 × 10^5^ per milliliter was in Δ*Mogap1*, which was significantly decreased compared with Guy11 after 9 days of culture on CM (Figure 2B). In order to determine whether MoGap1 was involved in conidial development, the septum and the nucleus of the conidia were stained using CFW and DAPI for observation, respectively. The results showed that no distinguishable differences were found between the three strains; all had a normal two-separated and three-nucleus structure (Figure 2C).

### 2.3. Increased Sensitivity to Rapamycin in MoGAP1-Deleted Mutants

Gap1 permeases activate the target of rapamycin complex 1 (TORC1) in yeast cells [23] and the latter plays an important role in controlling cell growth [24]. This led to the question of whether the yeast Gap1 homologue MoGap1 was sensitive to rapamycin. In order to answer the question, we simultaneously cultured the wild-type Guy11, Δ*Mogap1*, and Δ*Mogap1-C* on CM and a rapamycin medium (Rap). In the wild-type and the complementary strain Δ*Mogap1-C*, the relative inhibition rate was nearly 35%; the relative inhibition rate reached 45% in the Δ*Mogap1* mutant (Figure 3A,B). We then detected the growth rate of the wild-type Guy11 and Δ*Mogap1* strain in a liquid medium under microaerophilic conditions. The Δ*Mogap1* strain showed slightly slower growth compared with the Guy11 strain when cultured in the CM liquid medium for 2 days; the growth rate was obviously inhibited when treated with 100 ng/mL rapamycin (Figure 3C). The results showed that Δ*Mogap1* was more sensitive to rapamycin than the wild-type Guy11, indicating that the deletion of MoGap1 resulted in a decrease in the TOR activity. We then detected the virulence of Δ*Mogap1*. When the conidia of the wild-type Guy11 and Δ*Mogap1* were incubated on barley leaves for 4 days, lesions appeared and no differences were found between Guy11 and Δ*Mogap1*. However, the lesions were slightly restricted in the Δ*Mogap1* mutant compared with the wild-type when treated with 100 μg/mL rapamycin (Figure 3D).

### 2.4. MoGap1 Negatively Regulates Autophagy

The TOR protein kinase functions in two distinct multiprotein complexes, named TORC1 and TORC2 [25]. In nutrient-rich conditions, TORC1 becomes active, promotes the anabolic processes of proteins and lipids by integrating upstream signaling molecules (such as growth factors), and inhibits catabolic processes such as autophagy through the phosphorylation of its downstream effectors. In unfavorable environments (such as nitrogen starvation and TORC1 or TORC2 depletion), the general protein synthesis is downregulated and autophagy is upregulated. As the TOR protein kinase negatively regulates autophagy, the *MoGAP1* knockout mutant was sensitive to rapamycin and was presumed to reduce the TOR activity. In order to verify this speculation, we determined the TOR activity in Δ*Mogap1* by detecting the activity of the 40S ribosomal subunit (Rps6) by Western blotting. The activity of Rps6 is mainly regulated by the TOR kinase and the TOR activity can indirectly correspond with the phosphorylation level of Rps6. By adding rapamycin, the phosphorylation level of Rps6 in the wild-type decreased, indicating that rapamycin could inhibit the TOR activity in *M. oryzae*. However, the phosphorylation level of Rps6 in the mutant after adding rapamycin was much lower than that of the wild-type (Figure 4A,B), indicating that the TOR activity of Δ*Mogap1* was reduced.

Previous studies have found that autophagy in *M. oryzae* is regulated by TOR. In order to clarify the changes in the level of autophagy in Δ*Mogap1*, we used GFP-MoAtg8 (a GFP tag attached to the N-terminus of MoAtg8) to measure the autophagic flux. With the occurrence of autophagy, GFP-MoAtg8 is bound to the autophagosome membrane from the cytoplasm to the vacuole. Under the condition of nutrition starvation, MoAtg8 is rapidly degraded by proteases in the vacuole. With an increase in the level of autophagy, free GFP continuously accumulates; the level of free GFP can reflect the level of autophagy [26]. In our conidia, the punctate structures (autophagosomes) appeared both in the wild-type Guy11 and Δ*Mogap1*. However, the amount of autophagosomes in the wild-type were higher than in the Δ*Mogap1* strain. After 12 h of being induced, nearly 70% of the autophagosomes were degraded into the vacuole in the wild-type and almost all of the autophagosomes were degraded in the Δ*Mogap1* strain (Appendix A). Autophagy flux was then observed in the hypha. In the CM medium, the GFP fluorescence was mainly located in the cytoplasm and few of the punctate structures appeared in the wild-type. However, strong punctate structures were observed in the Δ*Mogap1* strain and many GFP were present in the vacuole. When we shifted the wild-type Guy11 and Δ*Mogap1* to an SD-N starvation medium for 4 h, punctate structures appeared in the wild-type whereas almost all the GFP-MoAtg8 was degraded into the vacuole (Figure 4C). Furthermore, the relative contents of GFP-MoAtg8 and GFP were detected by Western blotting. The results showed that the expression levels of GFP-MoAtg8 in Guy11 and Δ*Mogap1* gradually decreased with an increasing duration of starvation induction whereas the amount of free GFP gradually increased. In contrast, the degradation level of GFP in Δ*Mogap1* was significantly higher than that in Guy11, indicating that the level of autophagy in the mutant was higher than that in the wild-type (Figure 4D). Combined with the results concerning the expression of Rps6, our results showed that the deletion of MoGap1 reduced the TOR activity and increased the level of autophagy, indicating that MoGap1 negatively regulates autophagy.

### 2.5. MoGap1 Responds to Cell Wall Stress and Carbon Source Stress

The studies of recent years have reported that the TOR signaling pathway responds to cell wall stress and carbon source stress by regulating the Mps1 and AMPK signaling pathways [27,28,29]. In order to explore whether *MoGAP1* knockout mutants were involved in the regulation of these signaling pathways, we investigated the effects of hyperosmotic stress, cell wall stress, and starvation stress. Each of the three strains was cultured on a NaCl medium, KCl medium, SBT medium, SDS medium, CR medium, and SD-N medium, with each of them cultured on CM as a control. Their growth performance is shown in Figure 5A; Δ*Mogap1* was sensitive to CR and no significant changes were made by other stress factors (Figure 5A,B).

In order to further explore the factors that influenced the growth of the *MoGAP1* knockout mutants, we used minimal media (MM) as the control and replaced the C source used for the growth of the control with different nutrient sources such as oleic acid, mandelic acid, palmitic acid, and ferulic acid (Figure 6A). As shown in Figure 6B, except for ferulic acid, there were significant differences in the growth rates between Δ*Mogap1* and Guy11 under other C sources (Figure 6B).

## 3. Discussion

It is well-known that autophagy occurs under various environmental stresses such as nutrient deprivation, growth factor deficiencies, and hypoxia and it promotes the circulation of intracellular organic matter [30,31,32,33]. In *M. oryzae*, autophagy also affects the formation of its conidia and appressoria. The deletion of autophagy-related genes in *M. oryzae* can cause a loss of its pathogenicity [34,35,36,37,38,39]. Different regulatory pathways function in different stages of the formation, growth, and infection of *M. oryzae*, also affecting autophagy. Studies have shown that a few of the signal transduction pathways involved in the regulation of autophagy converge at TOR [40,41,42]. On the surface of lysosomes, amino acids (including leucine, arginine, and glutamine) are bound to the Ragulator-Rag complex, v-ATPase, SLC38A9, and KICSTOR to mediate mTORC1 activation [43]. mTORC1 can affect autophagy by regulating the ATG1/ULK1 complex, the DAP1 and TFE/MITF transcription factors, and p300-mediated acetylation [17]. On the other hand, mTORC2 has a bidirectional regulatory effect on autophagy [44]; it interacts with AKT to promote the phosphorylation of FOXO3 and inhibit the transcription of two autophagy-related genes (i.e., LC3 and BNIP3), thereby inhibiting the induction of autophagy. Additionally, mTORC2 plays an important role in the maturation and trafficking of autophagic vesicles [45]. It can be seen that autophagy-related and TOR-related genes are important in the pathogenic process of *M. oryzae*.

In yeast, Gap1 has a different mechanism from that of other specific transport systems. It is a general amino acid transporter whose substrates include acidic, neutral, and basic amino acids as well as L- and D-type isomers. Mutations at the Gap1 site are the only known mutations that specifically affect the general permeability of amino acids [46]. Springael et al. found that ammonium ions could induce Gap1 inactivation through Npi1/Rsp5 ubiquitin ligase (Ub) and degrade Gap1 in the vacuole [47]. Furthermore, Merhi et al. mutated different regions of Gap1 and revealed the importance of 64 regions in its secretion, transport, and downregulation [48]. Moreover, Van et al. studied the interaction between Gap1 proteins and found that Bsc6 and Yir014 could affect their transport and downregulation; the deletion of EGD2, YNL024c, or SPC2 inactivated Gap1 transport and signaling in a normal plasma membrane [49]. However, the functions of Gap1 in *M. oryzae* remain unclear. In this experiment, the principle of homologous recombination was used to knock out *MoGAP1*. The pathogenicity of Δ*Mogap1* on barley and rice was not changed with a conidial inoculation. However, the spore generation was significantly lower than that of the wild-type. As the function of Gap1 is closely related to TOR activity in yeast cells, in order to explore the relationship between them in *M. oryzae* we determined the TOR activity of Δ*Mogap1* by measuring MoRps6 and found that the deletion of MoGap1 resulted in a decrease in the TOR activity. As the TOR activity negatively regulated autophagy, the results indirectly suggested that the deletion of *MoGAP1* could enhance the autophagic ability. This conclusion was also confirmed by the determination of the activity of MoAtg8. Therefore, we believe that the *MoGAP1* gene negatively regulates autophagy in *M. oryzae*. Congo red (CR) transiently induced cell wall damage [50]; the inhibition rate of Δ*Mogap1* by CR was significantly higher than that of the wild-type, indicating that MoGap1 was involved in maintaining the CWI of *M. oryzae*. Through a comparative analysis, it could be seen that MoGap1 was conducive to the utilization of oleic acid, but hindered the absorption of mandelic acid and palmitic acid. Therefore, MoGap1 showed varied functions in the absorption and utilization of different carbon sources.

## 4. Conclusions

This study preliminarily explored the function of MoGap1 in *M. oryzae* through a phenotypic observation, a stress analysis, autophagy flux detection, and TOR activity monitoring. The loss of MoGap1 led to a conidia reduction as well as increased sensitivity to cell wall stress and carbon source stress in *M. oryzae*. In addition, the Δ*Mogap1* mutant displayed high sensitivity to rapamycin and showed a low TOR activity. Moreover, the level of autophagy significantly increased due to the low activity of TOR. In summary, our results suggested that MoGap1, as an upstream regulator of TOR signaling, regulated autophagy and responded to cell wall stress. The role of MoGap1 in *M. oryzae* remains to be further explored. There is still a long way to go in the exploration into the infection mechanism of *M. oryzae*.

## 5. Materials and Methods

### 5.1. Test Strains and Plants

The wild-type strain Guy11 of *M. oryzae* used in this experiment was from the N.J. Talbot laboratory in the United Kingdom. The plasmid vector was extracted from *Escherichia Coli* strain *DH5α*. The mediator of the *AtMT* method was Agrobacterium tumefaciens. The adopted plant was barley (*Hordeum vulgare* cv ZJ-8). The bacterial strains obtained in the experiment were stored in a laboratory-grade −80 °C refrigerator using a method of glycerol preservation. The fungal strains were stored in a laboratory-grade −20 °C refrigerator using the method of filter paper preservation.

### 5.2. Gene Knockout and Complementation

The gene knockout was conducted using an homologous recombination [19] in which the ~1000 bp of the upstream fragments of the target gene was amplified and inserted into the *PKO3A* vector together with the hygromycin phosphotransferase (HPH) gene. The knockout vector was transferred into the wild-type Guy11 strain using the *AtMT* method. In *M. oryzae*, previous studies verified the mutant deletion using a Southern Blot method [51]. In this study, we used another detection method as described by Lu et al. [22].

The complementation experiment was conducted in the *PKD5-GFP* vector by amplifying the *MoGAP1* sequence and inserting the sequence with the green fluorescent protein (GFP) gene into the vector before the complemented sample was transferred into the knockout strains and verified by observing the fluorescence of the transformants and by PCR.

For the mutant complementation, the upstream primer was designed based on the target gene and its upstream fragment of ~1000 bp; the downstream primer was designed based on the downstream fragment. The upstream and downstream fragments, the target gene, and the Bar resistance fragment with a flag tag were inserted into the *PKO3A* vector and then transferred into the knockout strains.

### 5.3. Pathogenicity Detection

For the barley infection test, Guy11, Δ*Mogap1*, and Δ*Mogap1-C* were cultured on CM for 7 days. The conidia were collected and diluted to a concentration of 1 × 10^5^ spores/mL, then incubated on 2-week-old barley with a conidial suspension and cultured at 25 °C for 4 days. Finally, the disease severity on the leaves was observed.

### 5.4. Cell Staining and Stress Test

In order to stain the septum and the nucleus of the conidia, the counted conidia (1 × 10^5^ conidia/mL) were dropped on a glass slide for 5 min and then the CFW dye solution was added to stain the samples for 3 min before they were observed under a fluorescence microscope with a UV laser. The conidial nuclei were stained with DAPI for 3–5 min. For drug stress, all strains were cultured on CM with 100 ng/mL rapamycin for 7 days. The growth performance of the colonies was observed and photographed and the diameter of each sample was measured to calculate the inhibition rate. The procedure was repeated 3 times for each strain. For the hyperosmotic stress test, CM with 0.7 M NaCl, 0.7 M KCl, and 1 M sorbitol were prepared. For the detection of cell wall stress, CM was used as the basal medium and 0.005% SDS media and 400 mg/mL Congo Red (CR) media were prepared. For the starvation stress test, SD-N media (yeast extract: 1.7 g/L, glucose: 20 g/L) were prepared. The inoculation steps were the same as those for the chemical stress. For the carbon source replacement, MM (with the peptone, casein extract, and yeast extract removed from CM) was used as the basal medium and 1% glucose C source was replaced by 1 mM oleic acid, 1 mM mandelic acid, 1 mM palmitic acid, and 1 mM ferulic acid, respectively. The inoculation steps were the same as those for the chemical stress.

### 5.5. Western Blot Analysis

For the TOR activity detection, 0.2 g of fresh mycelia was collected for culturing on CM for 7 days before 4 h of induction using 30 ng/mL rapamycin. The protein was extracted by a TCA/acetone extraction method [20] and the TOR activity was evaluated by the phosphorylation level of MoRps6. For the autophagic flux detection, the wild-type Guy11 and Δ*Mogap1* were cultured on liquid CM for 2 days and then starved for 4 h and 8 h on SD-N media before the whole proteins were extracted using a lysis buffer (50 mM Tris-HCl, 100 mM NaCl, 0.5 mM EDTA, 1% Triton X-100, 2 mM PMSF, 1% protease inhibitor, and 0.5% phosphatase inhibitor). The GFP-MoAtg8 and free GFP bands were detected using GFP antibodies.

## Figures and Tables

**Figure 1 ijms-23-13663-f001:**
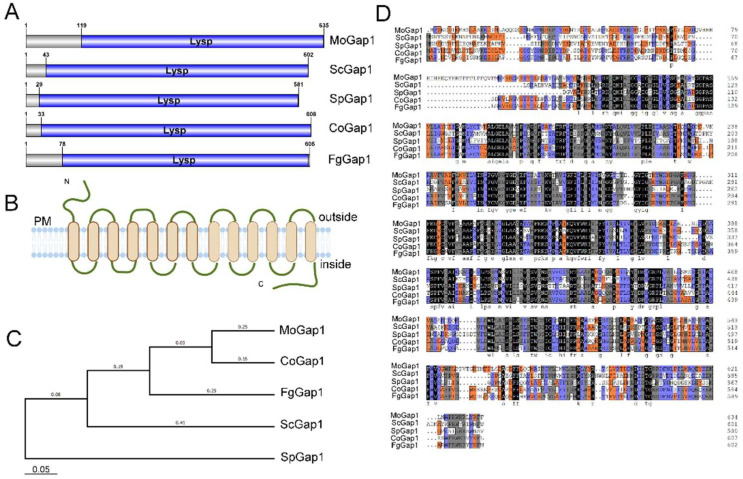
Identification of MoGap1 in *M. oryzae*. (**A**) Gap1 proteins of *M. oryzae*, *S. cerevisiae*, *S. pombe*, *C. truncatum*, and *F. graminearum* shown using IBS 1.0 software. Lysp: amino acid permease domain. (**B**) Schematic diagram of a Gap1 protein that contained 13 transmembrane regions. (**C**) Phylogenetic trees of the Gap1 proteins constructed using MEGA 7.0. (**D**) Multiple sequence alignment shown using DNAMAN software.

**Figure 2 ijms-23-13663-f002:**
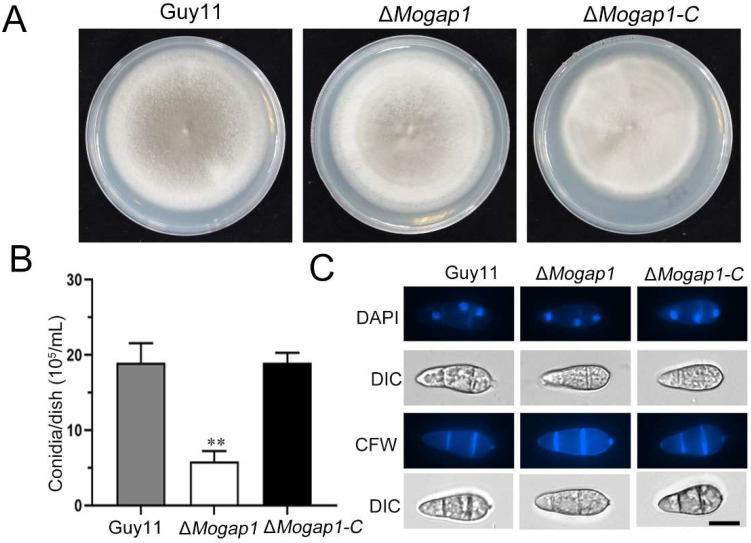
Colony morphology and sporulation of *M. oryzae*. (**A**) Colony morphology of Guy11, Δ*Mogap1*, and Δ*Mogap1-C* after 9 days of culture. (**B**) The conidia of Guy11, Δ*Mogap1*, and Δ*Mogap1-C* stained using DAPI and CFW on a scale of 10 μm. (**C**) Relative spore formation of Guy11, Δ*Mogap1*, and Δ*Mogap1-C* after 9 days of culture, the first of which differed significantly from the others. ** *p* < 0.01.

**Figure 3 ijms-23-13663-f003:**
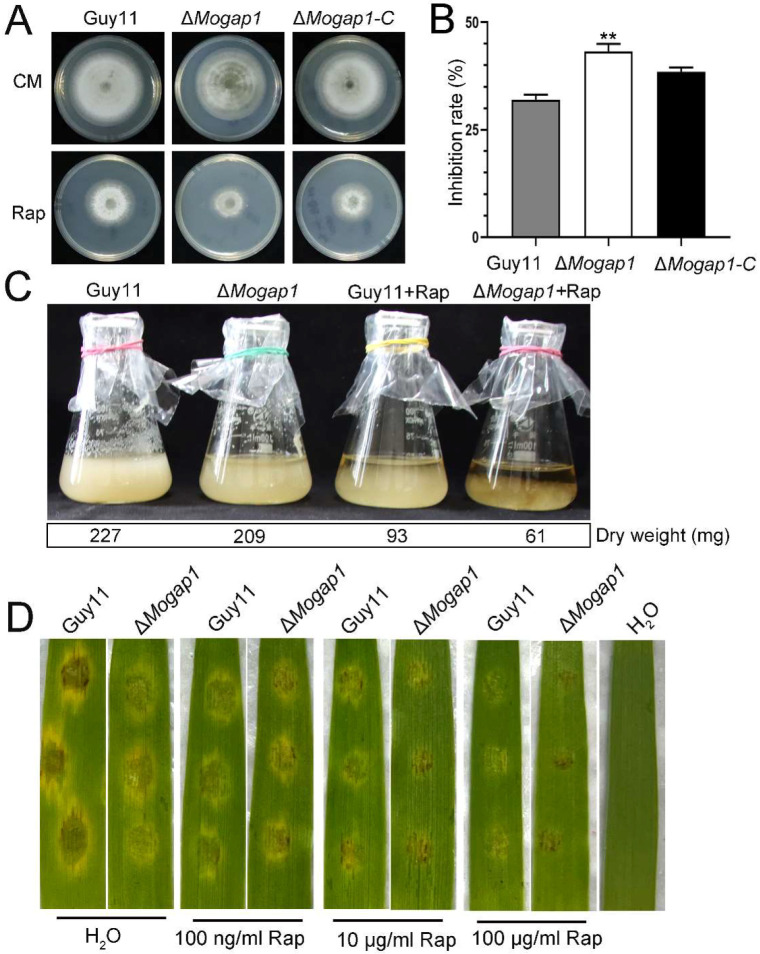
MoGap1 responses to rapamycin stress. (**A**) The growth of Guy11, Δ*Mogap1*, and Δ*Mogap1-C* for 8 days of culture on CM and Rap for 8 days, respectively. (**B**) The inhibition rates of rapamycin in Guy11, Δ*Mogap1*, and Δ*Mogap1-C* calculated by measuring the growth diameters, which were significantly different between MoGap1 and Guy11 (** *p* < 0.01). (**C**) The growth rate was observed in CM liquid medium. The dry weights of Guy11 and Δ*Mogap1* were calculated after being cultured in CM liquid medium for 2 days. (**D**) Virulence detection in Δ*Mogap1* mutant. The conidia (1 × 10^5^ spores/mL) were incubated on 2-week-old barley and cultured at 25 °C for 4 days.

**Figure 4 ijms-23-13663-f004:**
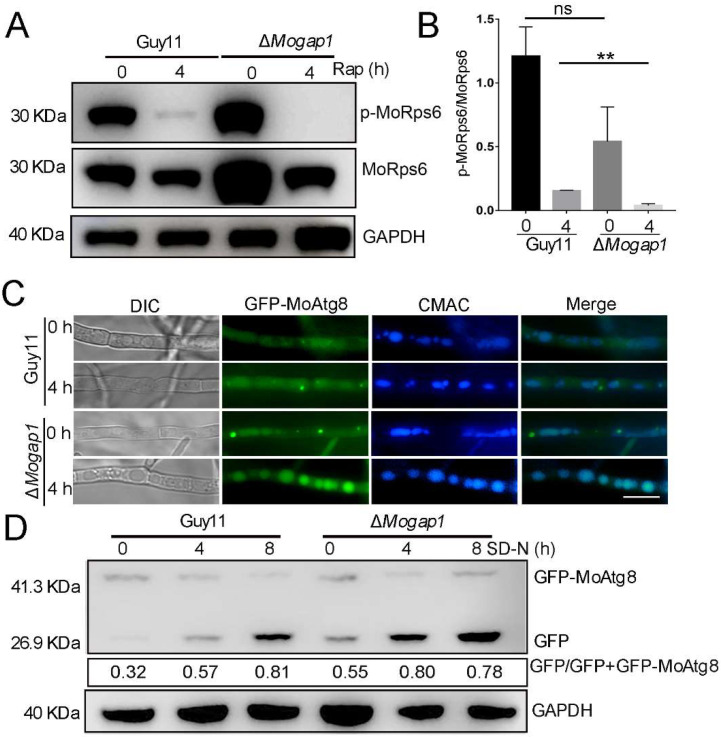
MoGap1 negatively regulates autophagy. (**A**) MoRps6 phosphorylation levels of Guy11 and Δ*Mogap1* after 0 h and 4 h of 30 ng/mL rapamycin induction with GAPDH as internal reference. (**B**) The phosphorylation levels of MoRps6 were calculated and graphed using Prism 7.0 software (*t*-test; ns: no significant, ** *p* < 0.01). (**C**) Autophagy flux was observed in the wild-type Guy11 and Δ*Mogap1* strain in CM and starvation conditions. Bar = 10 μm. (**D**) Expression levels of GFP-MoAtg8 and GFP in Guy11 and Δ*Mogap1*, respectively, after SD-N starvation with GAPDH as internal reference.

**Figure 5 ijms-23-13663-f005:**
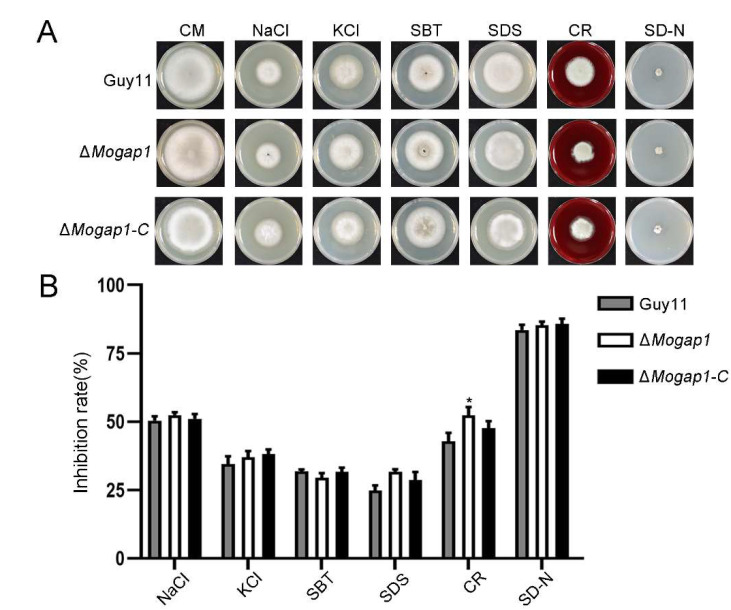
Effect of MoGap1 on CWI. (**A**) The growth performance of Guy11, Δ*Mogap1*, and Δ*Mogap1-C* under hypertonic stress (NaCl medium, KCl medium, and SBT medium), cell wall stress (SDS medium and CR medium), or starvation stress (SD-N medium). (**B**) The inhibition rates of Guy11, Δ*Mogap1*, and Δ*Mogap1-C* in different stress media, in which those between Δ*Mogap1* by CR and Guy11 were significantly different (* *p* < 0.05).

**Figure 6 ijms-23-13663-f006:**
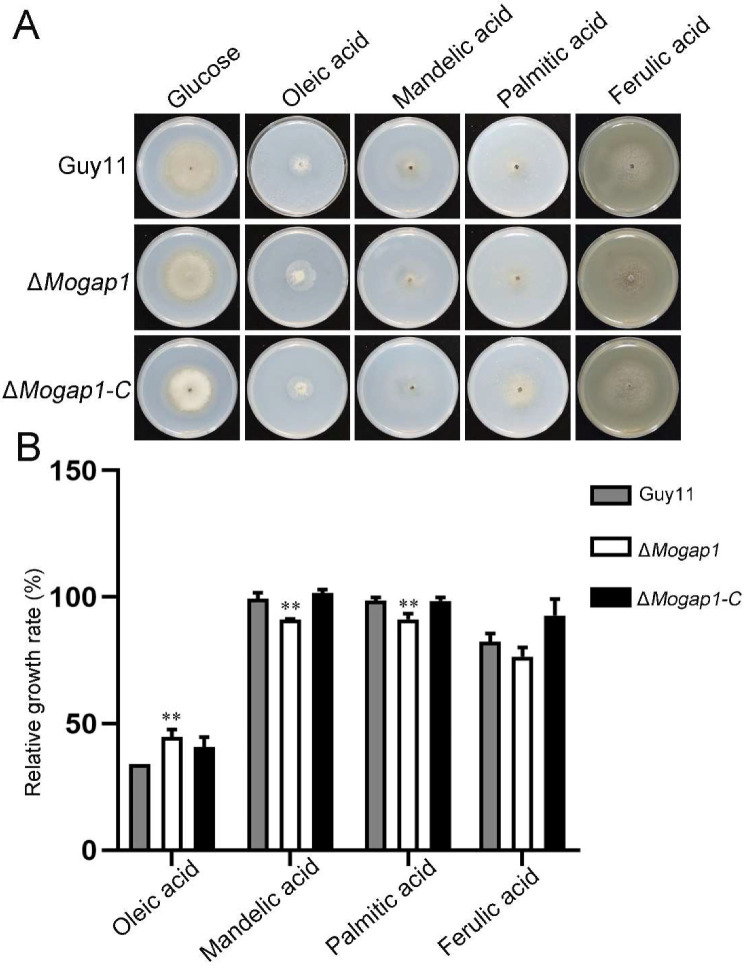
Utilization of different carbon sources in Δ*Mogap1*. (**A**) The growth of Guy11, Δ*Mogap1*, and Δ*Mogap1-C* with different C sources. (**B**) The relative growth rates of Guy11, Δ*Mogap1*, and Δ*Mogap1-C* with different C sources. With oleic acid, mandelic acid, or palmitic acid as the C source, the relative growth rates of Guy11 and Δ*Mogap1* were significantly different from each other. ** *p* < 0.01.

## Data Availability

Not applicable.

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
