# Peer review of "The Amino Acid Permease MoGap1 Regulates TOR Activity and Autophagy in *Magnaporthe oryzae"

_ijms, 2022, doi:10.3390/ijms232113663_

Round 1

Reviewer 1 Report

In this article, the investigation of the TOR activity in M. oryzae has been carried out. The introduction provides sufficient relevant references, the methods are appropriately described and the results are clearly presented. Apart from that, the manuscript is well written and the graphs are easy to understand.

Author Response

Thank you so much for your positive comments.

Reviewer 2 Report

The manuscript investigated the regulatory role of MoGap1 on vegetative growth, stress response, autophagy, TOR activity and virulence of M. oryzae based on identifying a yeast amino acid permease homologue MoGap1 in M. oryzae. The results suggested that MoGap1, as an upstream regulator of TOR signaling, regulates autophagy and responds to adversities such as cell wall stress by regulating TOR activity. Some suggestions have been made in order to improve the quality of this paper.

1. Authors have represented abstract in more generalized form. Authors should emphasize the levels of changes in different parameters assessed in % values.

2. In Materials and Methods section, the method of pathogenicity detection was described in 4.3, but the results was not found in Results section.

3. Fluorescence photo of the autophagic flux should be provided.

4. Conclusion or Schematic diagram of MoGap1 action should be provided.

5. The Latin name of fungi should be italic.

Author Response

Comments and Suggestions for Authors

The manuscript investigated the regulatory role of MoGap1 on vegetative growth, stress response, autophagy, TOR activity and virulence of M. oryzae based on identifying a yeast amino acid permease homologue MoGap1 in M. oryzae. The results suggested that MoGap1, as an upstream regulator of TOR signaling, regulates autophagy and responds to adversities such as cell wall stress by regulating TOR activity. Some suggestions have been made in order to improve the quality of this paper.

  1. Authors have represented abstract in more generalized form. Authors should emphasize the levels of changes in different parameters assessed in % values.

Re: Thank you for your suggestions. We added the parameters in this version.

  1. In the Materials and Methods section, the method of pathogenicity detection was described in 4.3, but the results were not found in the Results section.

Re: Thanks. We added the pathogenicity assays in the results section.

  1. Fluorescence photo of the autophagic flux should be provided.

Re: Thank you for your kind reminder. The fluorescence photo of the autophagic flux has been provided in this version.

  1. Conclusion or Schematic diagram of MoGap1 action should be provided.

Re: Thank you for your suggestion. We added the conclusion in the last paragraph.

  1. The Latin name of fungi should be italic.

Re: Thank you very much. We checked our manuscript and made modifications.

Reviewer 3 Report

In the manuscript ‘The amino acid permease MoGap1 regulates TOR activity and autophagy in Magnaporthe oryzae’, the authors performed the set of experiments, proving that M. oryzae Gap1 permease, responsible for amino acid uptake, is also involved in TOR-dependent signalling and autophagy. The authors created delta-Mogap1 mutant, which is impaired in proper conidiation and displays increased autophagy. In my opinion, presented work could be interesting to the other fungal researchers, especially working on pathogenic fungal species. The experiments gave the interesting results. Unfortunately, entire work lacks some fundamental experiments, required to be published. I have some points, which should improve the presented data.

Major points:

Lack of infection assay in Results, despite its description in section Materials and Methods

Lack of Southern Blot analysis to confirm gap1-deletion mutant (Southern Blot with Magnaporthe DNA is possible to do: DOI:10.3389/fpls.2017.01091)

Since cellular signalling is different when fungus grows on surface as compared to its growth in liquid medium under microaerophilic conditions (such as inside the plant cell) it is worthy to present the growth rate in liquid media, to compare with plate results.

Figure 5 & 6: inhibition rate and carbon utilization growth rate are almost indistinguishable from the wild type control. The range of change ~2-5% is almost insignificant.

Minor remarks:

Row 65: Atg1 chaperone Atg13

Row 70: in A. thaliana

Row 72: citation needed

Row 109: Amino acid...this sentence better move to introduction

Row 119: Using Tubulin...this sentence is unclear: how tubulin may verified mutant?

Row 122: knocked-out

Row 256: high-throughput...creation of one mutant is far from high-throughput

Row 296: For mutant complementation, rather than in situ

Row 312: please, rewrite this sentence into passive mode

Row 314: ...under fluorescence microscope

Row 323: yeast extract?

Author Response

Comments and Suggestions for Authors

In the manuscript ‘The amino acid permease MoGap1 regulates TOR activity and autophagy in Magnaporthe oryzae’, the authors performed the set of experiments, proving that M. oryzae Gap1 permease, responsible for amino acid uptake, is also involved in TOR-dependent signalling and autophagy. The authors created delta-Mogap1 mutant, which is impaired in proper conidiation and displays increased autophagy. In my opinion, presented work could be interesting to the other fungal researchers, especially working on pathogenic fungal species. The experiments gave the interesting results. Unfortunately, entire work lacks some fundamental experiments, required to be published. I have some points, which should improve the presented data.

Re: Thank you for your professional suggestions. We revised and added some fundamental experiments in the new manuscript.

Major points:

Lack of infection assay in Results, despite its description in section Materials and Methods。

Re: Thank you very much. We added the pathogenicity assays in the results section.

Lack of Southern Blot analysis to confirm gap1-deletion mutant (Southern Blot with Magnaporthe DNA is possible to do: DOI:10.3389/fpls.2017.01091)。

Re: Thank you for your professional ideas. Yes, Southern Blot analysis is essential for mutant detection. We added this reference in our manuscript (Zhang et al. 2017). In this version, we detected gap1-deletion mutant using double PCR methods combined with qPCR for the insertion copies of HPH in the mutant genome according to previously described by Lu et al. (Lu et al. 2014). And we added the detailed description in the materials and methods section and Figure S1.

(1) Zhang, X., Wang, G., Yang, C., Huang, J., Chen, X., Zhou, J., Li, G., Norvienyeku, J., & Wang, Z. (2017). A HOPS Protein, MoVps41, Is Crucially Important for Vacuolar Morphogenesis, Vegetative Growth, Reproduction and Virulence in Magnaporthe oryzae. Frontiers in plant science, 8, 1091. https://doi.org/10.3389/fpls.2017.01091

(2) Lu, J., Cao, H., Zhang, L., Huang, P., & Lin, F. (2014). Systematic analysis of Zn2Cys6 transcription factors required for development and pathogenicity by high-throughput gene knockout in the rice blast fungus. PLoS pathogens, 10(10), e1004432. https://doi.org/10.1371/journal.ppat.1004432

Since cellular signalling is different when fungus grows on surface as compared to its growth in liquid medium under microaerophilic conditions (such as inside the plant cell) it is worthy to present the growth rate in liquid media, to compare with plate results.

Re: Thank you for your insightful suggestions. We added this assay in results section (Fig S2).

Figure 5 & 6: inhibition rate and carbon utilization growth rate are almost indistinguishable from the wild type control. The range of change ~2-5% is almost insignificant.

Re: The data were repeated three times and the significant differences were tested by t-test. In Figure 5, there is a significant difference at the 95% confidence. And Figure 6 is a significant difference at the 99% confidence.

Minor remarks:

Row 65: Atg1 chaperone Atg13

Row 70: in A. thaliana

Row 72: citation needed

Row 109: Amino acid...this sentence better move to introduction

Row 119: Using Tubulin...this sentence is unclear: how tubulin may verified mutant?

Row 122: knocked-out

Row 256: high-throughput...creation of one mutant is far from high-throughput

Row 296: For mutant complementation, rather than in situ

Row 312: please, rewrite this sentence into passive mode

Row 314: ...under fluorescence microscope

Row 323: yeast extract?

Re: We revised these mistakes in our new version. Thank you.

Round 2

Reviewer 2 Report

All my suggestions Authors took into account. Now I recommend the article for publication.

Author Response

Thank you for your postive comments.

Reviewer 3 Report

The manuscript ‘The amino acid permease MoGap1 regulates TOR activity and autophagy in Magnaporthe oryzae’, corresponds to an updated version of a manuscript previously submitted to International Journal of Molecular Sciences. In this submission the authors have addressed few recommendations to the previous version of the manuscript. I have only few points to remark:

row 65: till unclear, this chaperone Atg1 or Atg13, or both of them?

row 79: under nitrogen...conditions

row 203: "we believe that the deletion of MoGap1 will reduce...", so the deletion has not been done? Yet it will be?

Figure 4a. How relevant are these results, these numbers? Whether it was replicated statistically from independent biological experiments?

In my opinion, the ATMT method with GFP cross-out is dubious and slippery, regarding lack of Southern confirmation, albeit people working with it and publish.

Author Response

Thank you for your comments and suggestions. We revised our manuscript again. Please see the attachment.
